# Ultra-clean and smoky marine boundary layers frequently occur in the same season over the southeast Atlantic

Sam Pennypacker[1], Michael Diamond[1], Robert Wood[1]

[1]Department of Atmospheric Sciences, University of Washington, Seattle, WA USA

*Correspondence to*: Sam Pennypacker (spenny@uw.edu)

**Abstract.** We study forty-one days with daily median surface accumulation mode aerosol particle concentrations below 50 cm$^{-3}$ (ultra-clean conditions) observed at Ascension Island (ASI; 7.9°S, 14.4°W) between June 2016 and October 2017 as part of the Layered Atlantic Smoke Interactions with Clouds (LASIC) campaign. Interestingly, these days occur during a period of great relevance for aerosol-cloud-radiation interactions, the southeast Atlantic (SEATL) biomass-burning season

(approximately June - October). That means that these critical months can feature both the highest surface aerosol numbers, from smoke intrusion into the marine boundary layer, as well as the lowest. While carbon monoxide and refractory black carbon concentrations on ultra-clean days do not approach those on days with heavy smoke, they also frequently exceed background concentrations calculated in the non-burning season from December 2016 - April 2017. This is evidence that even what become ultra-clean boundary layers can make contact with and entrain from an overlying SEATL smoke layer

before undergoing a process of rapid aerosol removal. Because many ultra-clean and polluted boundary layers observed at Ascension Island during the biomass burning season follow similar isobaric back-trajectories, the variability in this entrainment is likely more closely tied to the variability in the overlying smoke rather than large-scale horizontal circulation through the boundary layer. Since exceptionally low accumulation mode aerosol numbers at ASI do not necessarily indicate the relative lack of other trace pollutants, this suggests the importance of regional variations in what constitutes an 'ultra-

clean' marine boundary layer. Finally, surface drizzle rates, frequencies and accumulation – as well as retrievals of liquid water path – all consistently tend toward higher values on ultra-clean days. This implicates enhanced coalescence scavenging in low clouds as the key driver of ultra-clean events in the southeast Atlantic marine boundary layer. These enhancements occur against and are likely mediated by the backdrop of a seasonal increase in daily mean cloud fraction and daily median liquid water path over ASI, peaking in September and October in both LASIC years. Therefore the seasonality in ultra-clean

day occurrence seems directly linked to the seasonality in SEATL cloud properties. These results highlight the importance of two-way aerosol-cloud interactions in the region.

## 1 Introduction

Cloud-mediated aerosol radiative effects remain a significant source of uncertainty in our understanding of the climate system (Boucher et al., 2013; Rosenfeld et al., 2014). The Southeast Atlantic (SEATL) is a focal point for studying these

effects because biomass-burning aerosol (BBA) particles transported from central and southern Africa frequently overly a major stratocumulus deck between approximately July and October (Devasthale and Thomas, 2011; Zuidema et al., 2016c). The regional peak in satellite-retrieved cloud fraction and aerosol optical depth, as well as vertical overlap between the smoke layer and clouds, tends to occur between September and October (Adebiyi et al., 2015; Zuidema et al., 2016a). This establishes the potential for a complex web of aerosol-cloud-radiation interactions on seasonal and regional scales.

By absorbing solar radiation, BBA can alter the thermodynamic structure of the lower troposphere, leading to changes in low cloud cover (Gordon et al., 2018; Johnson et al., 2004; Sakaeda et al., 2011; Tummon et al., 2010; Yamaguchi et al., 2015; Zhou et al., 2017). If smoke entrains into the marine boundary layer (MBL) and activates into a cloud droplet, BBA may also induce indirect effects (Costantino and Bréon, 2013; Diamond et al., 2018; Zhou et al., 2017). However, contact between the base of smoke layers and the cloud-topped MBL is highly variable and difficult to constrain with satellite remote sensing (e.g. (Rajapakshe et al., 2017)). At Ascension Island (ASI, details below), there is frequently heavy smoke intrusion into the MBL earlier in the burning season (June-August) than expected given the later (September-October) peak in aerosol optical depth (Zuidema et al., 2018). We should note that ASI is situated further to the west of the 'classically' defined (Klein and Hartmann, 1993) SEATL stratocumulus region. The full role of BBA in the SEATL MBL particle budget and its subsequent interactions with low clouds remains under investigation.

Generally, aerosol particle number concentrations in the remote MBL exhibit significant spatio-temporal variability (Allen et al., 2011; Anderson et al., 2003; Mohrmann et al., 2017). One feature of this variability observed at the surface is periods of extremely low (ultra-clean) accumulation mode (approximately 100 nm – 1 μm) number concentrations (Pennypacker and Wood, 2017; Wood et al., 2017). Wood et al. (2017) noted relative enhancements in satellite retrievals of cloud liquid water path (LWP), a crucial driver of MBL coalescence scavenging (Wood, 2006), in ultra-clean air mass back trajectories several days before arriving over the Azores in the North Atlantic. Pennypacker and Wood (2017) further explored the properties of the post-frontal open cellular clouds typically associated with these ultra-clean conditions over the North Atlantic. Other studies have noted ultra-clean layers near the top of the MBL, in subtropical pockets of open cells and during the stratocumulus-to-cumulus transition (Petters et al., 2006; Terai et al., 2014; Wood et al., 2018). These examples from both mid-latitude and subtropical MBLs point to heavy drizzle-driven coalescence scavenging in regions of changing low cloud morphology as key for driving this particular feature of MBL aerosol variability. Drizzle also plays an important role in setting the mean MBL aerosol state under subtropical stratocumulus (Wood et al., 2012).

Our goal is to expand these analyses of ultra-clean conditions, as broadly defined for other regions in prior work noted above, into the SEATL, especially given the unique potential for influence from BBA. Our study is structured around the following three questions:

1. Do ultra-clean conditions occur at the surface in the SEATL, and what is their place in aerosol variability?

2. How do concentrations of biomass burning tracers during any ultra-clean conditions compare to background values from the non-burning season?

3. Are ultra-clean conditions associated with enhancements in precipitation?

To address these questions, we employ observations from the first Atmospheric Radiation Measurement (ARM) Mobile
Facility (AMF1) deployed to Ascension Island (7.9333°S, 14.41667°W) as part of the Layered Atlantic Smoke Interactions with Clouds (LASIC) campaign (Zuidema et al., 2016c, 2016b).

## 2 Data and Methods

### 2.1 Aerosol and Trace Gas Observations from AMF 1

A Droplet Measurement Technologies (DMT) Ultra-High Sensitivity Aerosol Spectrometer (UHSAS; Uin, 2016; DOI: 10.5439/1333828) provides aerosol number concentrations ($N_A$) at 0.1 Hz for particles with dry diameters between 60 nm and 1 μm. We define any day in the June 1, 2016 – October 30, 2017 LASIC UHSAS observational record (460 available days) as ultra-clean if the daily median $N_A$ falls below 50 cm$^{-3}$. UHSAS data is currently unavailable for July 2017 due to unresolved quality control issues. While admittedly somewhat subjective, this 50 cm$^{-3}$ threshold is consistent with the upper
bound of near-surface and below-cloud observations in MBL environments routinely featuring exceptionally low $N_A$ such as subtropical pockets of open cells (Abel et al., 2019; Sharon et al., 2006; Terai et al., 2014), mid-latitude open-cellular convection (Abel et al., 2017; Pennypacker and Wood, 2017) and across the trade wind stratocumulus-to-cumulus transition (Bretherton et al., 2019). It is also well situated within the typical range (~30 – 60 cm$^{-3}$) of number concentrations used for the lowest aerosol cases in large eddy simulation studies of MBL aerosol-cloud interactions (Wang et al., 2010; Wang and
Feingold, 2009; Yamaguchi and Feingold, 2015; Zhou et al., 2017). Prior work defined ultra-clean layers near the top of the MBL, often observed in the stratocumulus-to-cumulus transition, with $N_A < 10$ cm$^{-3}$ (Wood et al., 2018). We argue it is reasonable to set a higher threshold near the surface, where aerosol number concentrations are generally higher due to proximity to the sea spray source. Furthermore, Wood et al. (2018) focused on these layers primarily as a mesoscale feature within larger cloud systems, whereas our interest is in studying ultra-clean conditions as daily-scale events. Defining ultra-
clean conditions using daily median $N_A < 50$ cm$^{-3}$ balances the need to reasonably capture conditions with exceptionally low near-surface $N_A$ in the remote MBL while maintaining a robust sample of cases to study. We take daily medians as a better indication of the aerosol number concentration over the course of a day since they are more robust to any outlier observations than daily means, though this choice only leads to a discrepancy over one day identified as ultra-clean. Observations of total particle concentrations from a TSI Incorporated Ultrafine (>3 nm, $N_{CN3}$) Condensation Particle Counter
(CPC) compliment the UHSAS observations (Kuang, 2016; DOI: 10.5439/1046186).

We also consider measurements of carbon monoxide (CO) and refractory black carbon (rBC) mass concentrations from AMF1. CO concentrations are measured at 1 Hz by a Los Gatos Research trace gas analyser, while a DMT Single Particle Soot Photometer (SP2; (Sedlacek, 2017)) measures the rBC. The black carbon concentrations are calculated on 10-second intervals with a sensitivity of 10 ng m$^{-3}$. Our primary goal with these data (Question 2) is to determine whether ultra-clean

days represent the absence of any biomass burning influence in the MBL, relative to the regional background. This background is calculated from the non-burning season from December 2016 – April 2017. Both CO and black carbon act as biomass burning signatures, but since precipitation scavenging does not impact CO, it can reveal prior smoke contact even if aerosol concentrations are low. Again, we report median concentrations in order to minimize the potential impact of any outlier observations. We also report and compare inter-quartile ranges since a long tail on the distribution often skews the

variability about these medians.

## 2.2 Back Trajectories

Systematic differences in surface aerosol concentrations and composition at ASI, like those between ultra-clean and smoky days, could be explained by upwind differences in MBL entrainment from the free troposphere. The frequency of contact between the smoke base and MBL top over the SEATL is notoriously difficult to constrain because the aerosol often

significantly attenuates lidar beams (Rajapakshe et al., 2017). Zuidema et al. (2018) also posited that changes in transport pathway from the African continent, illuminated by three-dimensional back-trajectories, were key to explaining the smokiest conditions in the MBL near ASI. We take a complimentary approach by analysing 7-day isobaric boundary layer back-trajectories initialized at approximately 500 m over ASI at 12:00 UTC as computed by the NOAA Hybrid Single Particle Lagrangian Integrated Trajectory Model (HYSPLIT) with Global Data Assimilation System meteorology on a 0.5 degree by

0.5 degree grid (Stein et al., 2015). We compare the behaviour of these trajectories on ultra-clean days and days within the same months that exceed their monthly 90th percentile of daily median $N_A$, which we label as polluted. See Table S1 for a complete listing of the specific dates. Isobaric trajectories specifically reveal the origins and paths of the boundary layers that would be entraining smoke from the free troposphere.

## 2.3 Clouds and Precipitation


Based on prior analysis of ultra-clean conditions in the mid-latitude (Pennypacker and Wood, 2017; Wood et al., 2017) and subtropical (Petters et al., 2006; Wood et al., 2018) MBL, we hypothesize that enhanced drizzle also plays an important role in driving aerosol variability over the SEATL. Local surface precipitation rates at ASI are measured over a one minute averaging period using a Parsivel2 Laser Disdrometer (Delamere et al., 2016; DOI: 10.5439/1150252). We calculate a daily

precipitation frequency as the ratio of these averaging periods with a detected precipitation rate to the total within a day. This metric will of course not be a total drizzle frequency because it cannot include periods when precipitation evaporates before reaching the ground. We also examine differences in nonzero (i.e. only when clouds are detected) best-estimate retrievals of

liquid water path from the AMF 2-channel microwave radiometer (Cadeddu et al., 2013; Gaustad et al., 2016; DOI: 10.5439/1027369) between ultra-clean and all other days in the LASIC record. These retrievals are reported in 40 minute averaging windows. LWP is a key driver of MBL aerosol loss through coalescence scavenging even when drizzle doesn't reach the surface. In particular, we examine the statistics of retrieved LWP across bins of daily median $N_A$ (by 50 cm$^{-3}$ from

0 - 400 cm$^{-3}$, by 100 cm$^{-3}$ from 400 - 700 and then by 300 cm$^{-3}$ from 700 - 1000 cm$^{-3}$).  Finally, we place all of our observations in the context of the full LASIC record of both daily median MWR-retrieved LWP and cloud fraction as estimated by the ASI Total Sky Imager (Morris, 2005; DOI: 10.5439/1025308).

## 3 Results

### 3.1 Ultra-clean days

Forty-one days meet our criteria for ultra-clean conditions (daily median $N_A < 50$ cm$^{-3}$) in the available LASIC data. The 28 events from 2016 and the 13 events from 2017 all occur between July and November (Figure 1a, Table S1). The distribution

of events within these months varies, with August 2016 (12 days) and October 2017 (9 days) having the highest number of ultra-clean days in their respective years. As expected from Zuidema et al. (2018), we observe the highest daily $N_A$ peaks in the early 2016 and 2017 biomass burning seasons (June - August). Understanding SEATL MBL aerosol variability in this crucial period thus requires an understanding of both smoke intrusions and ultra-clean conditions. In months with few or none of these extremes (October 2016 - April 2017), the daily and monthly median particle concentrations vary more

consistently around 200 cm$^{-3}$.

Median $N_{CN3}$ mostly follows the same seasonal pattern as $N_A$ across the LASIC record (Figure 1b). This leads us to expect that the accumulation mode is generally an important driver of the variability in the total particle population. On ultra-clean days, however, the accumulation is by definition mostly depleted, while daily median $N_{CN3}$ ranges from a 115 cm$^{-3}$ minimum

to a 374 cm$^{-3}$ maximum. The range of total particle concentrations is therefore much higher than the range within the accumulation mode. On all except ultra-clean days, daily median $N_A$ explains more than half ($r^2 = 0.65$) of the variance in the total particle concentration, as expected (Figure 1c). This relationship is substantially weaker ($r^2 = 0.06$), with the 95% confidence interval for this correlation including zero, on ultra-clean days (Figure 1d). While ultra-clean days tend to have lower $N_{CN3}$ than other days, certainly those with smoke intrusions, the weakened correlation with $N_A$ further confirms that

different processes are responsible for governing the range of total particle concentrations outside of the accumulation mode. A similar difference in correlation strength between ultra-clean and other days holds at hourly time scales as well (Figure S1).

### 3.2 Biomass Burning Signatures

Perhaps unsurprisingly, CO generally tracks the accumulation mode aerosol number concentrations in Figure 1, correlating with daily median $N_A$ most strongly ($r^2 > 0.65$) in the early biomass burning seasons (June – August 2016 & 2017) when
smoke influence in the boundary layer is highest (Figure 2a). Outside of the primary burning season (December 2016 – April 2017), the day-to-day $N_A$-CO correlation strength varies with $r^2$ values between 0.04 and 0.49, depending on the month. rBC also generally follows the same patterns as aerosol number and CO (Figure 2b), with day-to-day $N_A$-rBC correlation again strongest ($r^2 > 0.55$) in the early burning season. The 2017 observations again confirm the analysis of Zuidema et al. (2018), which based on the 2016 data, found that the signature of smoke in the ASI MBL is strongest earlier in the traditional
SEATL biomass burning season. There is a smaller but noticeable peak in black carbon in January/February 2017 (Fig. 2b) that is oddly not evident in the CO observations. We leave a full diagnosis of this secondary peak for future work.

Of primary interest to this study is the range of BBA signature observations during ultra-clean events, relative to a background value. Prior observations in the subtropical Southern Hemisphere have put background CO concentrations
between 50 - 60 ppb (Allen et al., 2008, 2011; Shank et al., 2012). The median of hourly median CO concentration on ultra-clean days is 69 ppb, with an inter-quartile range of 62 - 74 ppb, and the full distribution of ultra-clean CO concentrations exhibits some moderate bi-modality (Figure 2c). In the non-burning season (December 2016 - April 2017), the distribution shifts to generally lower CO concentrations. The background median CO concentration is 59 ppb and the inter-quartile range is between 55 and 65 ppb, consistent with the prior estimates noted above. The first mode of ultra-clean CO concentrations
(Fig. 2c) overlaps more with the background distribution and is consistent with the background statistics. However, the second mode and longer tail of the distribution highlight the larger range of possible concentrations on ultra-clean days. This pulls the overall statistics toward higher concentrations on ultra-clean days relative to the non-burning background.

There is also some overlap in the distributions of ultra-clean and the non-burning background SP2 rBC (Dec. 2016, March-
April 2017, Figure 2d). However, as with CO, the statistics do indicate a shift toward overall higher concentrations on ultra-clean days. The median of hourly median SP2 rBC is 51 ng m$^{-3}$ with an inter-quartile range of 23 - 120 ng m$^{-3}$ on ultra-clean days, compared to the background median of 20 ng m$^{-3}$ and inter-quartile range of 12 - 45 ng m$^{-3}$. Even the hourly extremes captured by the 5th and 95th percentiles are higher on ultra-clean days (12 and 312 ng m$^{-3}$) than across the non-burning background (10 and 135 ng m$^{-3}$). In summary, there is no indication that ultra-clean days are devoid of BBA signatures or
even exhibit the same distribution of smoke tracer concentrations as the non-burning season background at ASI. We will return to the implication of these results for the characterization of extremely low aerosol number events as 'ultra-clean' in the Discussion.

Relative to the polluted extremes (recall these are defined by daily median $N_A$ above monthly 95th percentile), there are somewhat more ultra-clean boundary layer isobaric back-trajectories that originate farther toward the mid-latitudes and the Southern Ocean (Figure 3a). We might expect lower background aerosol concentrations and weaker influence from African biomass burning in these air masses than in those spending more time in the subtropics, helping explain the subset of ultra-clean days with burning tracer concentrations closer to background levels. However, trajectory latitude at seven days back from ASI only explains 25% of the variance in daily median CO concentrations across ultra-clean days. Trajectories from days with daily median CO ≤ 59 ppb (n = 6), the non-burning background median concentration, can be anywhere between 40° - 60°S at seven days back from ASI (Figure S2). Overall, boundary layer trajectory origin is a relatively weak predictor of downwind variability in CO concentration on ultra-clean days. Furthermore, there are many polluted and ultra-clean boundary layers that follow similar isobaric trajectories on their way toward ASI (Figure 3b). By three days away from ASI, most trajectories have converged to within three to four degrees latitude and longitude of each other. In other words, the boundary layers that would be entraining smoke from the tree troposphere often follow very similar horizontal circulation patterns for both the highest and lowest upstream extremes of $N_A$. This all points to a smaller role for variations in large-scale horizontal circulation in the SEATL MBL in driving aerosol and trace gas variability observed at ASI.

## 3.3 Precipitation and Cloud Liquid Water

Ultra-clean days exhibit markedly different surface precipitation characteristics, as measured by the ASI Parsivel2. The distribution of precipitation rates shifts toward higher intensities on ultra-clean days (Figure 4a). Precipitation is also much more common on ultra-clean days (Figure 4b), with almost 90% of non-UC days having a precipitation frequency of less than 0.05. The tendency for more frequent and more intense precipitation inevitably leads to higher total accumulation on ultra-clean days (Figure 4c). The difference mostly stems from the shift toward more frequent drizzle conditions in ultra-clean conditions. These data are all presented with cumulative distributions in order to concisely highlight the generally different behaviour of precipitation across ultra-clean days. However, the increase in drizzle intensity, frequency and accumulation also holds for ultra-clean days relative only to the distribution within their respective months (not shown).

The median LWP retrieved by MWR measurements is higher on ultra-clean days (110 g m$^{-2}$) compared to other days (76 g m$^{-2}$). The inter-quartile spreads are actually larger than the median LWP whether within ultra-clean days (41 – 235 g m$^{-2}$) or not (26 – 192 g m$^{-2}$). These statistics are further illustrated by the difference in the LWP cumulative distributions (Figure 5a). The shift is noted across most of the sampled range of LWP, though the distributions do overlap at the very highest values. While the shift toward higher LWP on ultra-clean days may not appear substantial, recall that coalescence scavenging is non-linearly dependent on LWP (Wood, 2006). The approximately 35% increase in median LWP on ultra-clean days would strengthen the coalescence scavenging aerosol sink by 70%.

Below a daily median $N_A$ of about 150 cm$^{-3}$, daily median LWP generally increases with decreasing $N_A$ (Figure 5b), still accompanied by high variability. This is indicative of higher LWP driving reductions in accumulation mode aerosol through drizzle production and scavenging. Over a wide range of intermediate daily median $N_A$ (~150 - 500 cm$^{-3}$) there is no discernible variation in binned LWP that would point to a dominant process. At higher number concentrations, however, daily median LWP continues to drop with increasing $N_A$, implicating at least some role for a relative lack of thick, drizzling clouds in sustaining the highest accumulation mode number concentrations.

## 4 Discussion and Conclusions

The SEATL remains the focus of intensive study because of the potential for direct, indirect and semi-direct radiative effects arising from extensive biomass burning aerosol layers overlying a major stratocumulus deck. We utilize data collected from an ARM Mobile Facility deployed (June 2016 - October 2017) to Ascension Island during the LASIC campaign to study 41 days with daily median accumulation mode aerosol concentrations below 50 cm$^{-3}$. Perhaps counter-intuitively, all of these observed ultra-clean days occur between July and November, the season where BBA concentrations in the SEATL region generally peak. In the 2016 observations, ultra-clean days are particularly prevalent in July and August, and frequently both precede and follow the periods of heavy smoke intrusion into the MBL around ASI examined in Zuidema et al. (2018). In 2017, most of the ultra-clean days occur in October, but we hesitate to comment on the robustness of any interannual variability given the relatively infrequent sampling of these events and the two-year observational record. Satellite retrievals of cloud droplet number concentration (Grosvenor et al., 2018) may provide a tool for extending our analysis with both a longer temporal record and greater spatial context of extreme depletion events in the SEATL MBL. However, these retrievals remain far more uncertain in the broken and/or heavily drizzling cloud scenes that often coincide with ultra-clean conditions. Nonetheless, both years of the LASIC deployment situate ultra-clean days as a feature of surface aerosol variability at ASI during the broader SEATL biomass burning season. This naturally leads to the question of what might drive this seasonality in the occurrence of ultra-clean days.

Surface precipitation rates, frequency and accumulation, as well as retrieved cloud LWP, are all systematically enhanced on ultra-clean days relative to non-ultra clean days. These observations are indicative of vigorous coalescence scavenging being a key driver of ultra-clean days at ASI. Clouds capable of initiating and sustaining this scavenging process are therefore likely precursors for these conditions. The months featuring ultra-clean days in the LASIC record are also the months leading up to and including the seasonal maximum in daily mean cloud fraction recorded by the ASI Total Sky Imager (Figure 6a). These same months also tend to be associated with the seasonal peak in daily median MWR best-estimate LWP values, though there is substantial spread in the monthly distributions in both 2016 and 2017 (Figure 6b). While limited to the two-year LASIC deployment period, these local observations are broadly consistent with prior work that has noted a seasonal maximum in satellite-retrieved SEATL regional cloud fraction (Zuidema et al., 2016a) and LWP (O'Dell et al.,

2008; Zuidema et al., 2016a) between August and October. And though these satellite-based analyses tend to consider data from an area to the southeast of ASI, scavenging upwind of our observations is also likely important. Thus, the seasonality in ultra-clean day occurrence appears broadly tied to the seasonality of SEATL clouds. The increase toward the seasonal maximum in cloud cover and LWP likely provides the necessary backdrop for enhancements in coalescence scavenging

needed to nearly fully deplete the accumulation mode in the MBL around ASI.

However, our results further shows that using the term "ultra-clean" incompletely describes conditions with extremely low accumulation mode particle number concentrations over the SEATL. Accumulation mode and total particle concentrations are generally well correlated at ASI, though much less so on ultra-clean days, on both daily and hourly scales. Even when $N_A$

$< 50$ cm$^{-3}$, smaller particles can have $2 - 4$ times the number concentrations than in the accumulation mode. The variability of Aitken and nucleation mode particle concentrations deserves more attention in future work, including any possibility of contributions from new particle formation. Carbon monoxide and refractory black carbon mass are also not necessarily at non-burning season (December – April) background levels despite the depletion of the accumulation mode. This points to the possibility of more frequent but subtler influence of smoke in the ASI MBL outside of the most extreme intrusions like

those examined by Zuidema et al. (2018). The wide range of trace pollutant concentrations observed over our sample of 41 days at ASI with exceptionally low $N_A$ highlights the importance of carefully considering what constitutes an 'ultra-clean' MBL in a particular region. More work is needed on systematically comparing the variability of pollutants like CO and rBC during periods of otherwise low accumulation mode aerosol number both within and between regions.

This analysis highlights an additional layer of complexity in the SEATL aerosol-cloud system. The months featuring the highest daily concentrations of aerosol particles in the MBL around Ascension Island also feature the lowest, likely due to multi-day time scale enhancements in coalescence scavenging on top of a pre-existing seasonal cycle. Ultra-clean MBL conditions present an important test for large eddy simulation (LES) physics and provide a tool for further probing underlying processes and their associated timescales. The initiation, evolution and persistence of these conditions could

make particularly interesting case studies for LES modelling of the Lagrangian evolution of the SEATL MBL. More broadly, air mass history is an important factor in the interpretation of aerosol-cloud interactions over the SEATL given the typical time scales associated with entrainment of free tropospheric aerosol into the MBL and loss from precipitation (Diamond et al., 2018). The broad similarities in isobaric boundary layer back trajectories even between ultra-clean and the most polluted days at ASI suggest that systematic differences in large-scale horizontal circulation in the boundary layer may

play less of a role in downwind (e.g. at ASI) aerosol variability. Instead, the vertical separation between smoke and the MBL along air mass trajectories, in addition to the co-evolution of clouds and precipitation, could set a balance between entrainment and scavenging. The transport and consequent three dimensional structure of BBA layers certainly varies with circulation patterns above the boundary layer (Adebiyi and Zuidema, 2016). Abel et al. (2019) also noted relative reductions in the entrainment of overlying smoke tracers into the MBL in a pocket of heavily drizzling open cells near ASI, potentially

driven by cloud dynamical differences noted in other previous work (Berner et al., 2011). The progression of clouds and precipitation along trajectories in the SEATL will depend on a number of factors, including potential influence of overlying smoke layers (Yamaguchi et al., 2015; Zhou et al., 2017) and buffering feedbacks (Stevens and Feingold, 2009). The detailed evolution of how all of this might lead to downwind ultra-clean conditions and the variations in other trace pollutant concentrations observed during these events should be further explored in a Lagrangian LES framework. Coarser resolution models used to study aerosol-cloud interactions across the broader SEATL region should also test their capability of reproducing these events and their place in the aerosol, cloud and meteorological seasonal cycles.

**Acknowledgements**

Sam Pennypacker's work was supported by the United States Department of Energy (DOE) award DE-SC0013489 (ENA Site Science). Michael Diamond's work was supported by NASA headquarters under the NASA Earth and Space Science Fellowship Program, grant NNX-80NSSC17K0404. Robert Wood's work was supported by NASA Earth Venture Suborbital-2 grant NNX-15AF98G. The U.S. Department of Energy, Office of Science, Office of Biological and Environmental Research, Climate and Environmental Sciences Division sponsors the Atmospheric Radiation Measurement (ARM) program. We are indebted to the scientists and staff who make these data possible by taking and quality controlling the measurements. The authors gratefully acknowledge the NOAA Air Resources Laboratory (ARL) for the provision of the HYSPLIT transport and dispersion model and/or READY website (http://www.ready.noaa.gov) used in this publication. Particular thanks to Paquita Zuidema for helpful discussions about the LASIC campaign and to Arthur Sedlacek for providing access to and discussing the SP2 dataset.

**Data Availability**

All LASIC ARM data are publicly available at https://www.archive.arm.gov/discovery/. ARM dataset DOI references are included in the text.

**Author Contributions**

SP and RW conceived the study design and analysis. SP performed the analysis of the LASIC data. MD ran the HYSPLIT transport and dispersion model to produce the back-trajectories. SP wrote the paper and all authors reviewed the manuscript.

**Competing interests**

The authors declare that they have no conflict of interest.

**Special Issue Statement**

This article is part of the special issue "New observations and related modelling studies of the aerosol–cloud–climate system in the Southeast Atlantic and southern Africa regions (ACP/AMT inter-journal SI)". It is not associated with a conference.

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

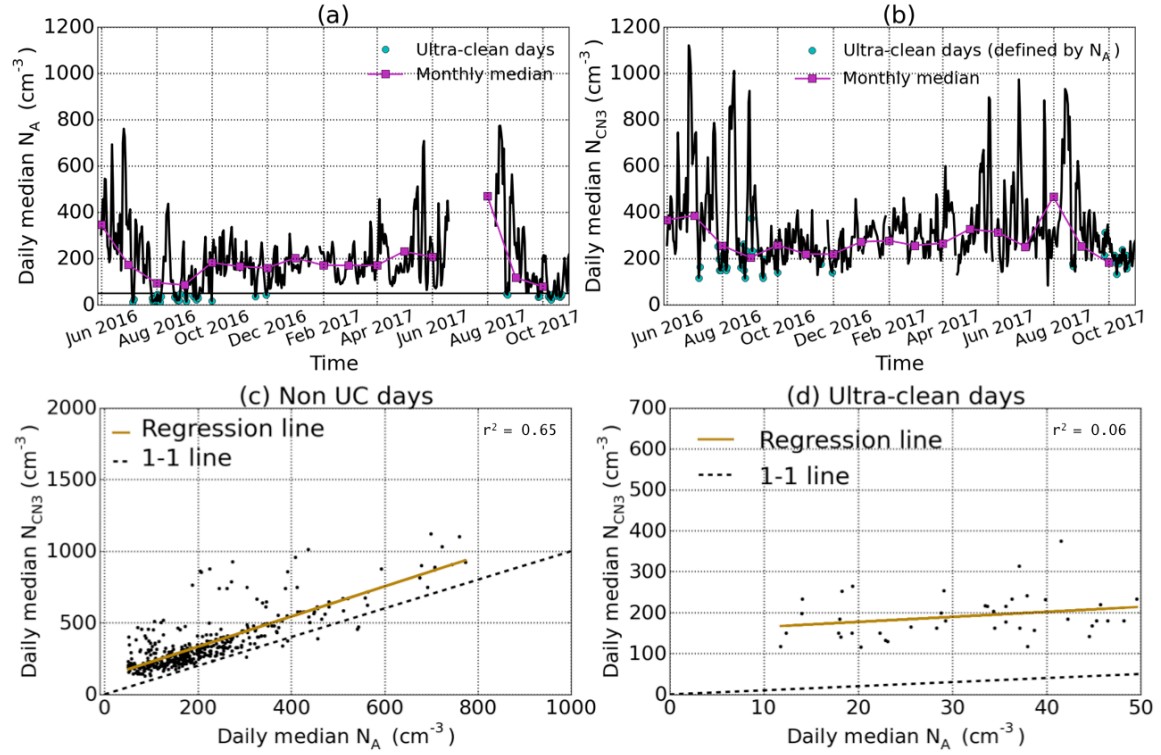

**Figure 1:** Time series of daily and monthly median (a) $N_A$ and (b) $N_{CN3}$ measured during LASIC, with ultra-clean days marked in cyan. In (a) and (b), vertical grid lines mark the first of each month labeled on the tick. Daily median $N_{CN3}$ is then regressed against daily median $N_A$ for both (c) non ultra-clean days and (d) ultra-clean days.

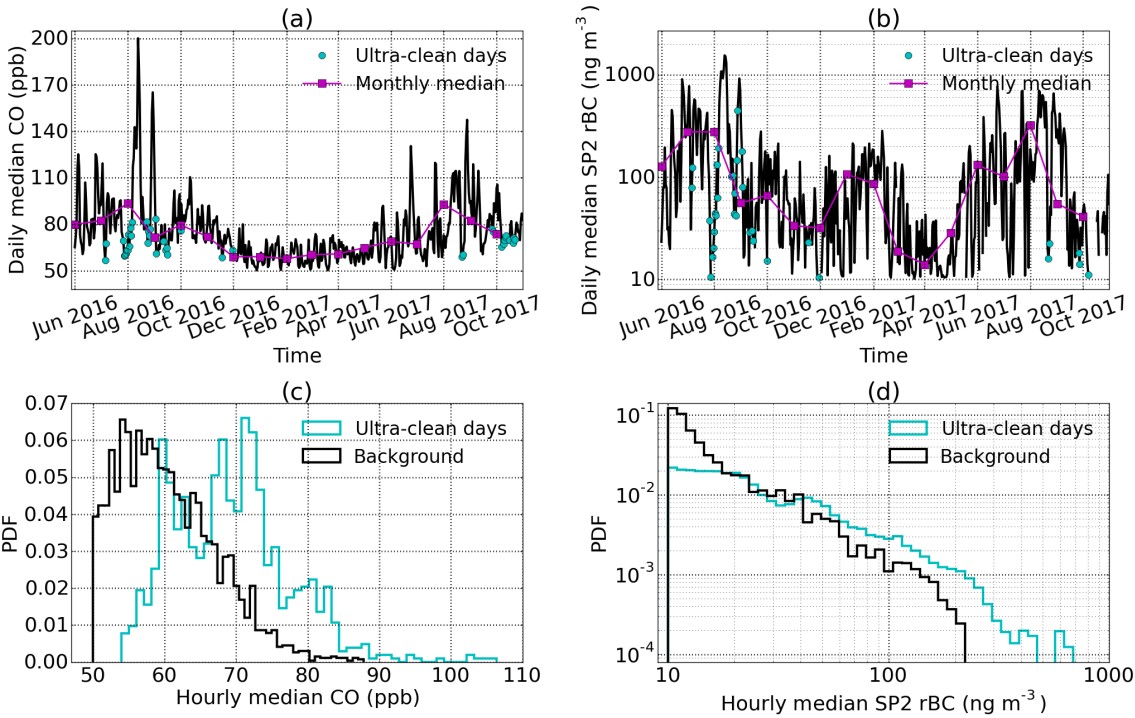

**Figure 2:** Time series of daily and monthly median (a) CO and (b) rBC measured during LASIC, with ultra-clean days marked in cyan. In (a) and (b), vertical grid lines mark the first of each month labelled on the tick. We then compare the PDF of hourly median (c) CO and (d) rBC from ultra-clean days to the PDF of hourly median concentrations from each tracers' respective non-burning background.

## Isobaric Back Trajectories

### (a) Ultra-clean, boundary layer

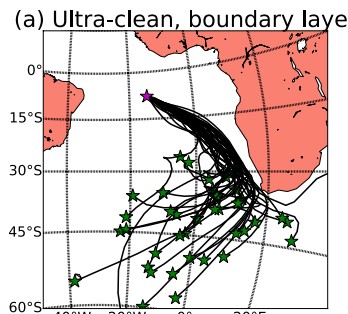

### (b) Polluted, boundary layer

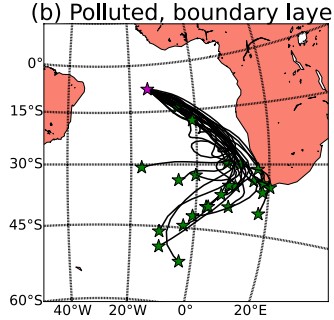

**Figure 3:** Isobaric 7-day HYSPLIT b                    00 m for (a) ultra-clean and (b) polluted

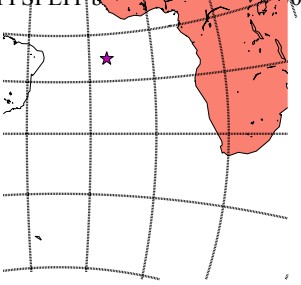

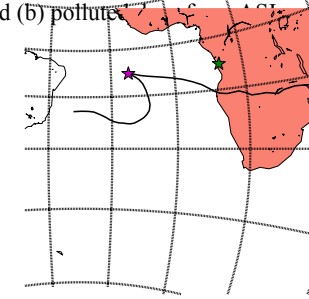

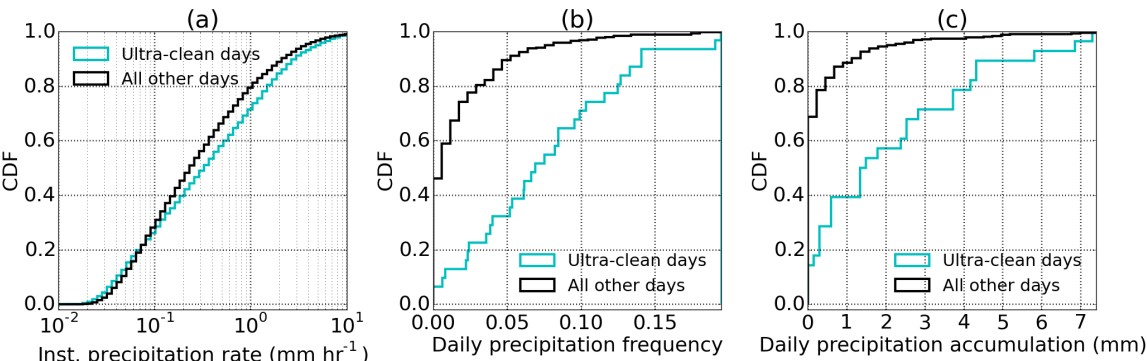

**Figure 4:** Cumulative distributions of (a) instantaneous precipitation rate, (b) daily precipitation frequency and (c) daily precipitation accumulation as measured by the ASI Parsivel2 laser disdrometer.

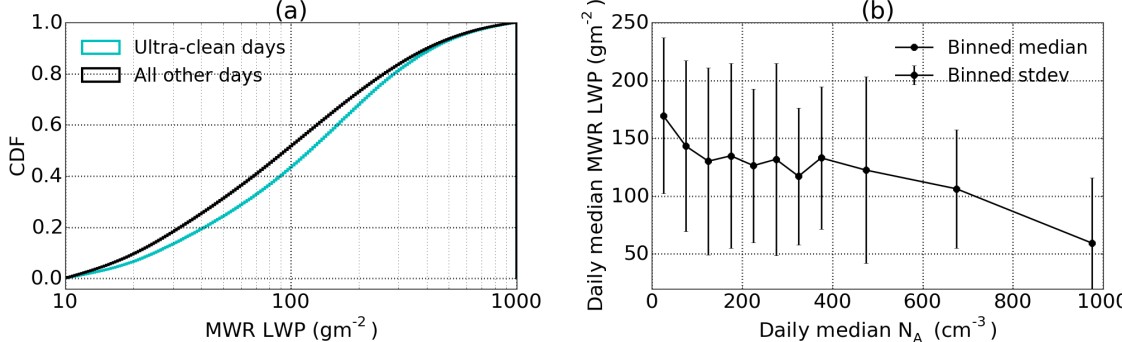

**Figure 5:** (a) Comparison of the cumulative distributions of best-estimate LWP retrieval from the ASI MWR between ultra-clean and all other days and (b) medians/standard deviations of daily median MWR LWP across bins of daily median accumulation mode aerosol for the entire LASIC record. In (b), bin widths were selected to account for varying density of days across the range of aerosol concentrations while still visualizing the broader pattern.

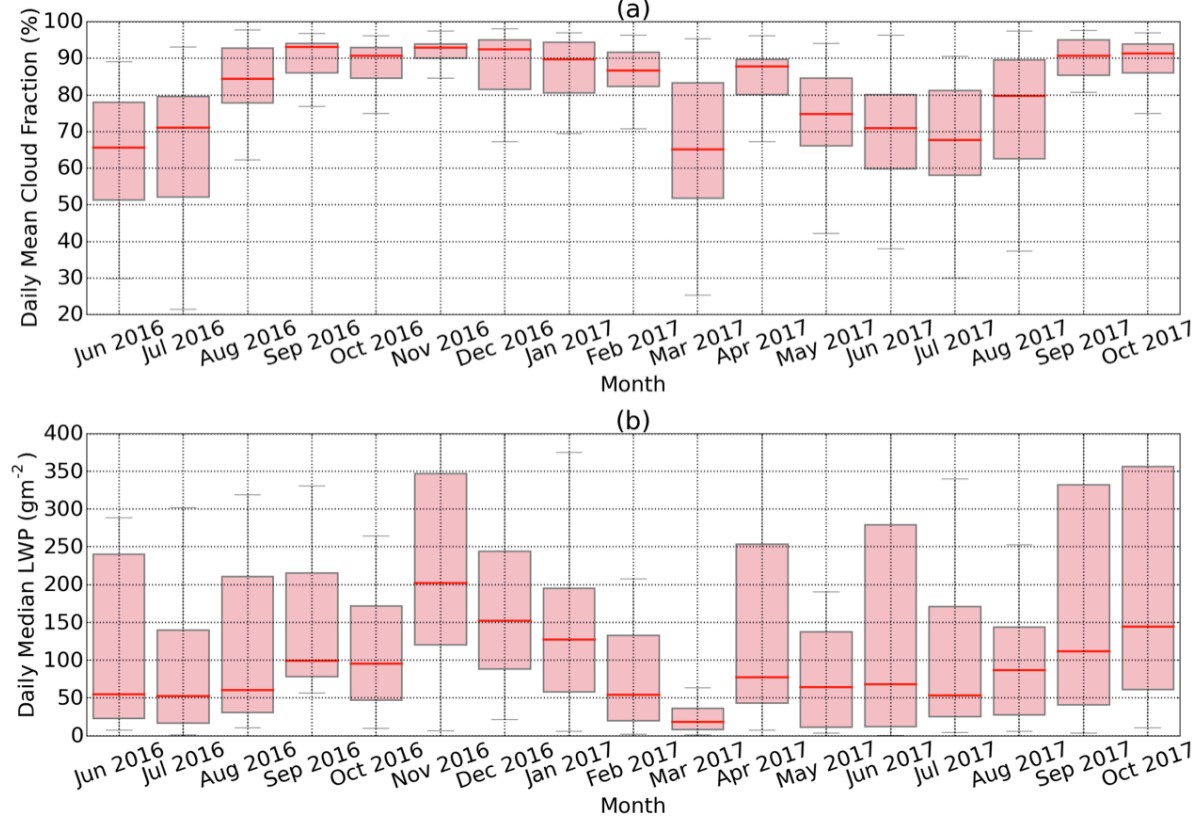

**Figure 6:** Monthly boxplots of (a) daily mean Total Sky Imager cloud fraction and (b) daily median MWR best-estimate LWP for each month in the LASIC record.