# Peer review of "Ultra-clean and smoky marine boundary layers frequently occur in the same season over the southeast Atlantic"

_Atmospheric Chemistry and Physics, 2019_

## Referee Comment (RC1) · Anonymous Referee #1 · 6 Aug 2019

acp-2019-628 Ultra-clean and smoky marine boundary layers frequently occur in the same season over the southeast Atlantic Sam Pennypacker, Michael Diamond, Robert Wood

This manuscript uses date from LASIC campaign and data from Ascension Island between June 2016 and October 2017. They catalogue ultra-clean events (41 in number) over this time period and compare them to smoky cases and background conditions. They find that there is evidence for seasonality in the occurrence of the ultra-clean layers and that they occur during the same time as smoky conditions. They propose mechanisms and ways that these ultra-clean days are related to cloud properties and

precipitation. The most interesting aspect of this work is the comparison between the smoky and ultra-clean days and how it related to drizzle amount and occurrence. This work is well written, easy to read and to follow, the results follow clearly from their analysis and the figures are well chosen and presented. My recommendation to accept his work with minor revisions

Main comments: 1) It would be helpful to justify why the authors chose < 50 cm-3 as ultra-clean. Is this arbitrary or is there some other reason for this choice?

2) The record is quite short. Is there any way to extrapolate information about ultra-clean days from satellite data sets to get an idea of how often these ultra-clean days occur in a longer-term record? This could be brought up in the discussion section, perhaps, as future work.

3) For the figures, the tick marks are hard to read and the numbers bleed into the figure space. Also, for Figure 2 there should be a few more markers (or at least tick marks) on (d).

4) For Figure 2, what do the PDFs for non-UC days look like for CO and rBC from the SP2? It'd be interesting to see the comparison for the non-clean days in these PDFs.

No Line by Line Comments: This work was well edited and has no issues in the text.

---

## Referee Comment (RC2) · Anonymous Referee #2 · 10 Aug 2019

Review of "Ultra-clean and smoky marine boundary layers frequently occur in the same season over the southeast Atlantic" by Sam Pennypacker et al.

This manuscript presents a really interesting observation using data from a recent field campaign (LASIC) in the southeast Atlantic, such that both the highest and lowest accumulation mode aerosol particle concentrations have been observed in the same season, i.e. the biomass burning season, highlighting the high daily variability of MBL aerosol loading and the level of complexity between aerosol-cloud interactions in the region. In the study, comparisons between the ultra-clean days and the background non-burning days are made, in terms of CO and BC concentrations, LWP, and pre-

cipitation intensity and frequency, to highlight the role of coalescence scavenging in creating these ultra-clean days in between highly polluted days.

I find the manuscript well-written and easy to follow, with well-organized sections and clear questions to address.

Major comments:

1. Medians with inter-quartiles are used throughout the manuscript, favored over means with standard deviations. I wonder if there is a reason for such preference, and suggest specifying why medians better represent the characteristics of these variables at the daily time scale?

2. The polluted days used for the back-trajectory comparison raised a concern. As most of the ultra-clean days are from the biomass burning season, would it make more sense to compare directly to the polluted days within the BB season, i.e. excluding the non-burning season? Besides, even the upper 5 percentile of daily median Na from December to April aren't really "polluted," are they?

Minor comments:

P2-13, only one set of parentheses is needed here.

P3-9, Is there a reason or any reference suggesting 50 as the threshold for ultra-clean condition? What is the instrument sensitivity of the UHSAS measurements?

P3-14, I suggest including the sampling frequency as well as the instrument sensitivity for CO and rBC measurements.

P3-28, the full name of HYSPLIT should be introduced here, and details on the HYSPLIT runs should be given as well, e.g. what meteorological dataset is used, at what spatial resolution.

Section 2.3, sampling frequency of the laser disdrometer and the microwave radiometer should be given here, and I assume the 2-channel MWR is the one at the AMF1

site, not the one at the airport, correct?

P4-24, "available LASIC data," I am curious to know how many available days there are in total, i.e. 41 out of how many days?

P5-5∼7, could you label the $r^2$ values on Fig. 1 c and d?

P5-10, could you provide $r^2$ values for the hourly correlations in Fig. S1 (on the figure as well)?

P5-16, this sentence is unclear to me, are you saying $r^2>0.65$ is for the early biomass burning season, which is a subset of your data? Please define the early biomass burning season, i.e. which month?

P5-17, "day-to-day the correlation" → "the day-to-day correlation", and you are talking about CO to Na correlation here, right?

P5-18, "rBC generally follows . . ." suggest adding $r^2$ values here.

P5-22, agreed, for future work, maybe check with the SP2 instrument mentor, as SP2 is an optical device, and things that are not smoke can still trigger it.

P5-26, "median hourly median", how about "median of hourly median"?

P5-27, I suggest putting these statistics on the figure as well, maybe in the same color as the ultra-clean days, but in dashed lines, this will help me a lot to visualize the shift.

P6-9∼10, see the second comment in Major comments, and how many polluted days are there, based on your criteria?

P6-25, I agree with you on the use of CDF, but I find PDFs are useful to see as well, have you considered showing both of them on the same plot? Well, this could make a really busy plot, I will leave the decision up to you.

P6-29, Why the mean is shown here instead of median, what are the median values of these?

Figure 4 a, suggest adding those vertical dashed lines in the background when log scale is used, just like you did in figure 5 a.

Section 4, In the discussion, you proposed reasons for the fact that these ultra-clean days are prone to appear during the BB season. Besides the fact that the seasonal peaks in LWP and CF coincide with this time period, I think of the buffering system introduced by Stevens and Feingold, Nature, 2009, i.e. high loading of Na -> strong indirect effect -> higher LWP -> strong scavenging -> remove Na. As you mentioned a two-way aerosol-cloud interactions in the abstract, have you considered this as another possibility?

---

## Referee Comment (RC3) · Anonymous Referee #3 · 26 Aug 2019

This is an interesting and mostly clearly-written manuscript. However, I was startled to learn that 'ultra-clean', as the authors have defined it, can still apply to boundary layers with elevated CO and rBC concentrations. It invites the question: what is a boundary layer with low Na, and non-elevated CO and rBC concentrations? 'extreme-clean' ? I suggest that the authors rephrase 'ultra-clean' as 'ultra-low-aerosol' (or something along those lines), to be more specific. The term 'ultra-clean' appears to have been defined from northern hemisphere studies in which this distinction was not relevant, but I think for a new reader that the term 'ultra-clean' is confusing. The authors themselves touch on this on p. 8, second paragraph.

[Figure]

p. 2, line ~18: this is where the term 'ultra-clean' is introduced, in parentheses, with the paragraph providing detail on the prior studies that have used this term. I suggest including a subsequent paragraph that discusses how this term may or may not apply well to the southeast Atlantic, and use that to define 'ultra-low-aerosol'.

P. 3, line 9: definition of 'ultra-clean' ('ultra-low-aerosol') needs more justification. Likely this follows that in prior studies, given the importance of this definition would suggest mentioning the definition within that paragraph on p.2 (and using a different term).

P.3 line 30: Would high aerosol counts but low CO/rBC qualify as 'polluted' ? The authors suggest this might occur during February. Overall a bit more description of the high-Na days would be helpful. Are they all from the months when smoke is clearly present?

Section 3.2, fig. 3: It would be interesting to also discriminate further those days that are more truly pristine. Do those correspond to the back-trajectories that more clearly go back to the southern oceans? There may not be many days with daily median CO values <~ 60ppb and rBC values within the sensitivity limit, but there should be some, and it would add interest to hear about those as well.

P. 6 line 22: what is the precipitation frequency on the UC days?

Figures:

Fig. 1: It's hard to tell how many UC days occur per month from panel a and b. One idea would be to mention how many occur each month near the top of the figure.

Fig. 2 panels c and d: I suppose this is saying something about temporal variability as well, with hourly values being shown for a given daily median threshold on Na. For completeness it would be nice to see a similar plot for the pdf of the hourly median Na. It would be a fifth panel. Not sure what to suggest for a 6th panel to balance it visually.

Fig. 4: the cumulative distributions take some study to interpret. Have the authors considered a normalized frequency distribution instead? Same for Fig. 5a.

Fig. 6: I don't see a clear correspondence between LWP and UC days through this figure. I wonder if the MWR LWP data are simply too local.

[Figure]

---

## Referee Comment (RC4) · Anonymous Referee #4 · 3 Sep 2019

**Manuscript summary**

The authors analyze surface observations of aerosol, gas phase composition, and cloud properties at Ascension Island over a period of 16 months, acquired during the Layered Atlantic Smoke Interactions with Clouds (LASIC) campaign. Back-trajectory calculations support the analysis. The authors distinguish three aerosol states at Ascension Island: Background conditions, polluted conditions, and ultraclean conditions. Ultraclean conditions are defined based on a daily median concentration of aerosol particles (CCN) with dry diameters between 60 nm and 1 $\mu$m < 50 cm$^{-3}$. The authors find 41 days with ultraclean conditions at Ascension Island. All of these occur dur-

ing the South-West African biomass burning season. A portion of the ultraclean days also exhibits carbon monoxide and refractory black carbon levels above background. Apart from ultraclean days, boundary layer CCN concentrations at Ascension Island are significantly elevated above background levels. No days with ultraclean conditions are found outside the biomass burning season, which defines background conditions. The authors conclude, based on analysis of carbon monoxide and refractory black carbon levels, statistics of precipitation and liquid water path at Ascension Island, and back-trajectory calculations that CCN concentrations are low on the ultraclean days not because originally clean air has been advected to Ascension Island, but because enhanced coalescence scavenging in low clouds has strongly reduced CCN in polluted air masses. This is an interesting result because it points to a more complex interaction between (anthropogenic) aerosol and cloud properties in the region, with causal links in both directions.

**Review summary**

In their analysis of the observations the authors accumulated a good amount of circumstantial evidence to render their hypothesis plausible, although the analyzed data are specific to conditions at Ascension Island only and hence do not establish a causal connection between conditions on ultraclean days and processes that may give rise to them. Although not quantitative, the back-trajectory analysis is helpful. The study is, as the authors point out in their closing statements, a good motivation and starting point for subsequent investigations.

There are a few points that I would ask the authors to look after, listed below. Otherwise, the manuscript is in good shape.

**Major comments**

- Could there be other explanations for the ultraclean days than enhanced coalescence scavenging in low clouds with higher liquid water content? E.g., is it possible that on the ultraclean days, the polluted air has entrained earlier into the boundary layer, hence spent a more time there compared to other days during the biomass burning season? A longer sojourn in the boundary layer would give coalescence scavenging more time to deplete the aerosol. Please comment and if applicable, discuss in the manuscript.

- Please calculate the average speed of the trajectories between 35 S and Ascension Island. Is there a difference in advection velocity between the ultraclean and non-ultraclean days during the biomass burning season? If yes, discuss what this could mean for the processes that cause ultraclean conditions.

- The criterion for what makes ultraclean conditions varies between works. Albrecht et al. (doi:10.1175/BAMS-D-17-0180.1), e.g., define ultraclean conditions as having aerosol concentrations of less than 10 cm$^{-3}$ in the nominal range between 0.06 - 1.0 $\mu$m., while in the present work it is < 50 cm$^{-3}$. Please add a passage mentioning the different criteria and explain why in the present work the criterion of < 50 cm$^{-3}$ was chosen.

- How robust is the number of ultraclean days to the UHSAS < 50 cm$^{-3}$ criterion?

- "... with the correlation statistically indistinguishable from zero ($r^2$ = 0.06), ..."

To make this statement you /must/ calculate the p-value of the linear regression/correlation. Without the p-value, there is no way of telling whether a correlation coefficient/coefficient of determination is statistically indistinguishable from zero, regardless of its numerical value.

**Other comments**

- Please check the text for sentences that can be simplified; some are hard to understand. For example,

"The relative invariance of isobaric boundary layer back trajectories between ultra-clean and the most polluted days at ASI suggests that the potential for BBA entrainment set by the vertical separation of a smoke layer and the evolution of the boundary layer cloud field plays a larger role in upwind (e.g. at ASI) aerosol variability than a systematic difference in large-scale horizontal circulation in the boundary layer."

is rather difficult to decipher.

- Please mention the meteorological input that you used to drive the HYSPLIT model.

- 500 m trajectories are not isobaric.

- Please consider if the labeling of the abscissa in the plots that show data as a function of the month is precise enough to inform the reader on the actual point in time (are the vertical lines the 1st of the month or the 15th?)

---

## Author Response (AR1)

**Author Response to Reviewers**

We thank all of the reviewers for their helpful comments on our manuscript. Our response to each of these points, including any relevant revisions, is detailed below. Referee comments are written in black font color, while the author responses are written in red. Any page and line numbers in our response refer to the revised manuscript.

**Referee #1**

This manuscript uses date from LASIC campaign and data from Ascension Island between June 2016 and October 2017. They catalogue ultra-clean events (41 in number) over this time period and compare them to smoky cases and background conditions. They find that there is evidence for seasonality in the occurrence of the ultra-clean layers and that they occur during the same time as smoky conditions. They propose mechanisms and ways that these ultra-clean days are related to cloud properties and precipitation. The most interesting aspect of this work is the comparison between the smoky and ultra-clean days and how it related to drizzle amount and occurrence. This work is well written, easy to read and to follow, the results follow clearly from their analysis and the figures are well chosen and presented. My recommendation to accept his work with minor revisions

1) It would be helpful to justify why the authors chose < 50 cm-3 as ultra-clean. Is this arbitrary or is there some other reason for this choice?

In response to this and other similar reviewer comments, we have added the following clarification to the text (P3, L15-26): "While admittedly somewhat subjective, this 50 $cm^{-3}$ threshold is consistent with the upper bound of near-surface and below-cloud observations in MBL environments routinely featuring exceptionally low $N_A$ such as subtropical pockets of open cells (Abel et al., 2019; Sharon et al., 2006; Terai et al., 2014), mid-latitude open-cellular convection (Abel et al., 2017; Pennypacker and Wood, 2017) and across the trade wind stratocumulus-to-cumulus transition (Bretherton et al., 2019). It is also well situated within the typical range (~30 – 60 $cm^{-3}$) of number concentrations used for the lowest aerosol cases in large eddy simulation studies of MBL aerosol-cloud interactions (Wang et al., 2010; Wang and Feingold, 2009; Yamaguchi and Feingold, 2015; Zhou et al., 2017). Prior work defined ultra-clean layers near the top of the MBL, often observed in the stratocumulus-to-cumulus transition, with $N_A < 10$ $cm^{-3}$ (Wood et al., 2018). We argue it is reasonable to set a higher threshold near the surface, where aerosol number concentrations are generally higher due to proximity to the sea spray source. Furthermore, Wood et al. (2018) focused on these layers primarily as a mesoscale feature within larger cloud systems, whereas our interest is in studying ultra-clean conditions as daily-scale events. Defining ultra-clean conditions using daily median $N_A < 50$ $cm^{-3}$ balances the need to reasonably capture conditions with exceptionally low near-surface $N_A$ in the remote MBL while maintaining a robust sample of cases to study."

2) The record is quite short. Is there any way to extrapolate information about ultraclean days from satellite data sets to get an idea of how often these ultra-clean days occur in a longer-term record? This could be brought up in the discussion section, perhaps, as future work.

We have added a brief mention of this point to our Discussion (P8, L16-20):

"Satellite retrievals of cloud droplet number concentration (Grosvenor et al., 2018) may provide a tool for extending our analysis with both a longer temporal record and greater spatial context of extreme depletion events in the SEATL MBL. However, these retrievals remain far more uncertain in the broken and/or heavily drizzling cloud scenes that often coincide with ultra-clean conditions."

3) For the figures, the tick marks are hard to read and the numbers bleed into the figure space. Also, for Figure 2 there should be a few more markers (or at least tick marks) on (d).
Thank you for suggesting changes to make sure our figures are clear. We have increased the thickness of the grid lines on our plots, added grid lines for the minor axis ticks in Figure 2d and increased the padding for the axis tick labels to avoid any bleeding.

4) For Figure 2, what do the PDFs for non-UC days look like for CO and rBC from the SP2? It'd be interesting to see the comparison for the non-clean days in these PDFs.
We have updated Figure 2 to include the PDFs for the non-burning background rather than just presenting the statistics.

**Referee #2**

This manuscript presents a really interesting observation using data from a recent field campaign (LASIC) in the southeast Atlantic, such that both the highest and lowest accumulation mode aerosol particle concentrations have been observed in the same season, i.e. the biomass burning season, highlighting the high daily variability of MBL aerosol loading and the level of complexity between aerosol-cloud interactions in the region. In the study, comparisons between the ultra-clean days and the background non-burning days are made, in terms of CO and BC concentrations, LWP, and precipitation intensity and frequency, to highlight the role of coalescence scavenging in creating these ultra-clean days in between highly polluted days. I find the manuscript well-written and easy to follow, with well-organized sections and clear questions to address.

Major comments:

1. Medians with inter-quartiles are used throughout the manuscript, favored over means with standard deviations. I wonder if there is a reason for such preference, and suggest specifying why medians better represent the characteristics of these variables at the daily time scale?
Thank you for your question about this choice. This decision is primarily motivated by the fact that means will generally be more sensitive to being biased by any outlier observations than medians. Furthermore, medians and inter-quartile ranges will also generally be more robust and illustrative (or at least as illustrative) if the distribution of observations is skewed (e.g. Figs 2c,d). We have updated the text in several locations to specifically address this:
- In reference to the use of daily median accumulation mode number concentration to identify ultra-clean events, we have added (P3, L26-28) the following clarification: "We take daily medians as a better indication of the aerosol number concentration over the course of a day since they are more robust to any outlier observations than daily means, though this choice only leads to a discrepancy over one day identified as ultra-clean."
- In reference to CO and rBC concentrations (P4, L8-10): "Again, we report median concentrations in order to minimize the potential impact of any outlier observations. We also report and compare inter-quartile ranges since a long tail often skews the variability about these medians."

2. The polluted days used for the back-trajectory comparison raised a concern. As most of the ultra-clean days are from the biomass burning season, would it make more sense to compare directly to the polluted days within the BB season, i.e. excluding the non-burning season? Besides, even the upper 5 percentile of daily median Na from December to April aren't really "polluted," are they?
Thank you for raising the need to clarify this point. The polluted back trajectories are only from the months where there are also ultra-clean days (i.e. during the biomass burning season). We have added text in the Abstract (P1, L16) and Section 2.2 (P4, L20-21), where the back trajectories are introduced, to ensure that this is clear. We have also added a supplemental table (Table S1) listing all of the ultra-clean and polluted dates from those same months that are included in the analysis.

*Minor comments:*

P2-13, only one set of parentheses is needed here.
Thank you, this has been fixed.

P3-9, Is there a reason or any reference suggesting 50 as the threshold for ultra-clean condition? What is the instrument sensitivity of the UHSAS measurements?
In response to this and other similar reviewer comments, we have added the following clarification to the text (P3, L15-26): "While admittedly somewhat subjective, this 50 cm$^{-3}$ threshold is consistent with the upper bound of near-surface and below-cloud observations in MBL environments routinely featuring exceptionally low $N_A$ such as subtropical pockets of open cells (Abel et al., 2019; Sharon et al., 2006; Terai et al., 2014), mid-latitude open-cellular convection (Abel et al., 2017; Pennypacker and Wood, 2017) and across the trade wind stratocumulus-to-cumulus transition (Bretherton et al., 2019). It is also well situated within the typical range ($\sim$30 – 60 cm$^{-3}$) of number concentrations used for the lowest aerosol cases in large eddy simulation studies of MBL aerosol-cloud interactions (Wang et al., 2010; Wang and Feingold, 2009; Yamaguchi and Feingold, 2015; Zhou et al., 2017). Prior work defined ultra-clean layers near the top of the MBL, often observed in the stratocumulus-to-cumulus transition, with $N_A < 10$ cm$^{-3}$ (Wood et al., 2018). We argue it is reasonable to set a higher threshold near the surface, where aerosol number concentrations are generally higher due to proximity to the sea spray source. Furthermore, Wood et al. (2018) focused on these layers primarily as a mesoscale feature within larger cloud systems, whereas our interest is in studying ultra-clean conditions as daily-scale events. Defining ultra-clean conditions using daily median $N_A < 50$ cm$^{-3}$ balances the need to reasonably capture conditions with exceptionally low near-surface $N_A$ in the remote MBL while maintaining a robust sample of cases to study."

P3-14, I suggest including the sampling frequency as well as the instrument sensitivity for CO and rBC measurements.
We have added (P4, L2) the sampling frequency for the CO measurements (1 Hz). According to the ARM documentation for the CO gas analyzer measurements, sensitivity is "not meaningful" for the CO measurements because tropospheric background values are "well above" minimum detectable levels. The sampling frequency and sensitivity of the SP2 rBC measurements were included in the original manuscript.

P3-28, the full name of HYSPLIT should be introduced here, and details on the HYSPLIT runs should be given as well, e.g. what meteorological dataset is used, at what spatial resolution.
The text has been updated to address these points: "We take a complimentary approach by analysing 7-day isobaric boundary layer back-trajectories initialized at approximately 500 m over ASI at 12:00 UTC as computed by the NOAA Hybrid Single Particle Lagrangian Integrated Trajectory Model (HYSPLIT) with Global Data Assimilation System meteorology on a 0.5 degree by 0.5 degree grid (Stein et al., 2015)."

Section 2.3, sampling frequency of the laser disdrometer and the microwave radiometer should be given here, and I assume the 2-channel MWR is the one at the AMF1 C2, not the one at the airport, correct?

The text has been updated (P5, L1-3) to include the relevant sampling/averaging windows, as well as to indicate that all data is from the AMF site (not the airport).

P4-24, "available LASIC data," I am curious to know how many available days there are in total, i.e. 41 out of how many days?

There are 460 available days of UHSAS data between June 1 2016 and October 30 2017. We have added a note of this in the text (P3, L12).

P5-5~7, could you label the rˆ2 values on Fig. 1 c and d?

Done.

P5-10, could you provide rˆ2 values for the hourly correlations in Fig. S1 (on the figure as well)?

Done.

P5-16, this sentence is unclear to me, are you saying rˆ2>0.65 is for the early biomass burning season, which is a subset of your data? Please define the early biomass burning season, i.e. which month?

Thank you for noting the need to clarify this. We now reference which months these statistics refer to (P6, L4-5).

P5-17, "day-to-day the correlation" → "the day-to-day correlation", and you are talking about CO to Na correlation here, right?

The text has been clarified to read: "the day-to-day $N_A$-CO correlation."

P5-18, "rBC generally follows . . ." suggest adding rˆ2 values here.

Done.

P5-22, agreed, for future work, maybe check with the SP2 instrument mentor, as SP2 is an optical device, and things that are not smoke can still trigger it.

We agree. This was outside the scope of our current analysis, but hopefully will get addressed in future work.

P5-26, "median hourly median", how about "median of hourly median"?

Thank you for the suggestion. Fixed.

P5-27, I suggest putting these statistics on the figure as well, maybe in the same color as the ultra-clean days, but in dashed lines, this will help me a lot to visualize the shift.

In response to the comments from Reviewer One, we decided to just show the background PDFs instead of plotting just the statistics. This will also hopefully help visualize the shift.

P6-9~10, see the second comment in Major comments, and how many polluted days are there, based on your criteria?

See response to the Major Comment 2 above.

P6-25, I agree with you on the use of CDF, but I find PDFs are useful to see as well, have you considered showing both of them on the same plot? Well, this could make a really busy plot, I will leave the decision up to you.

We appreciate that both data presentations can be useful, but we agree that having both on the same plot could be confusing. Our main goal here is to highlight the general differences and shifts in the distributions, which the CDF plots nicely demonstrate. To us, they also more clearly represent the parts of the distribution that contribute most to those shifts and by how much.

P6-29, Why the mean is shown here instead of median, what are the median values of these?

Thank you for pointing out this inconsistency. We now report the median and inter-quartile spread for each (P7, L27-29). This does not change any results qualitatively, but the means are higher than the medians by about $60 - 70$ $gm^{-2}$. This correction actually further highlights why we generally elected to report medians in this analysis (see response to Major Comment 1).

Figure 4 a, suggest adding those vertical dashed lines in the background when log scale is used, just like you did in figure 5 a.

Thank you for the suggestion, we have added those axis lines to Figure 4a.

Section 4, In the discussion, you proposed reasons for the fact that these ultra-clean days are prone to appear during the BB season. Besides the fact that the seasonal peaks in LWP and CF coincide with this time period, I think of the buffering system introduced by Stevens and Feingold, Nature, 2009, i.e. high loading of Na -> strong indirect effect -> higher LWP -> strong scavenging -> remove Na. As you mentioned a two-way aerosol-cloud interactions in the abstract, have you considered this as another possibility?

Our observations point to the importance of enhanced coalescence scavenging for driving and maintaining ultra-clean conditions in the SEATL MBL, but cannot indicate what might have caused the initial enhancement. However, given the evidence in our analysis for previous air mass contact with smoke even with low $N_A$, we certainly agree that the mechanism described in Stevens and Feingold (2009) may be at work at some point in the evolution of these boundary layers. The seasonal cycles of cloud cover and LWP in the region still seem as likely backdrops for generating these enhancements to enough of a point where extreme aerosol number depletions can develop. We have added mention in the Discussion of the Stevens and Feingold (2009) feedback as something for possible consideration in the Lagrangian LES studies we hope our results will motivate (P10, L3).

**Referee #3**

This is an interesting and mostly clearly-written manuscript. However, I was startled to learn that 'ultra-clean', as the authors have defined it, can still apply to boundary layers with elevated CO and rBC concentrations. It invites the question: what is a boundary layer with low Na, and non-elevated CO and rBC concentrations? 'extreme-clean'? I suggest that the authors rephrase 'ultra-clean' as 'ultra-low-aerosol' (or something along those lines), to be more specific. The term 'ultra-clean' appears to have been defined from northern hemisphere studies in which this distinction was not relevant, but I think for a new reader that the term 'ultra-clean' is confusing. The authors themselves touch on this on p. 8, second paragraph.

p. 2, line ~18: this is where the term 'ultra-clean' is introduced, in parentheses, with the paragraph providing detail on the prior studies that have used this term. I suggest including a subsequent paragraph that discusses how this term may or may not apply well to the southeast Atlantic, and use that to define 'ultra-low-aerosol'.

Thank you for your comments. We agree that the potential for influence from overlying smoke makes the SEATL a unique environment relative to many of the other regions where 'ultra-clean' conditions have been studied before. In fact, that's why we decided to extend analysis of these conditions into this region and shaped how we framed our three guiding research questions from the start. We feel that it is reasonable, at least for this study, to continue using the terminology that is consistent with the literature so far in order to directly connect to that prior work. We have made a few changes to the text to address this point:

- In the Abstract, we have added a new sentence (P1, L18-20) that more specifically addresses this potential discrepancy we had already mentioned in our Discussion and you raise in your comment: "Since exceptionally low accumulation mode aerosol numbers at ASI do not necessarily indicate the relative lack of other trace pollutants, this suggests the importance of regional variations in what constitutes an 'ultra-clean' marine boundary layer."
- We added "…as broadly defined for other regions in prior work noted above" when introducing our goal of expanding considerations of ultra-clean conditions to the SEATL at the end of the Introduction (P2, L31-32).
- We have added a sentence in Section 3.2 noting that we will return to the implications of the CO/rBC observations for 'ultra-clean' characterization in the Discussion (P6, L30-32).
- Added further emphasis on this distinction and the need for future work on this in the Discussion (P9, L15-18): "The wide range of trace pollutant concentrations observed over our sample of 41 days at ASI with exceptionally low $N_A$ highlights the importance of carefully considering what constitutes an 'ultra-clean' MBL in a particular region. More work is needed on systematically comparing the variability of pollutants like CO and rBC during periods of otherwise low accumulation mode aerosol number both within and between regions"

P. 3, line 9: definition of 'ultra-clean' ('ultra-low-aerosol') needs more justification. Likely this follows that in prior studies, given the importance of this definition would suggest mentioning the definition within that paragraph on p.2 (and using a different term).

In response to this and other similar reviewer comments, we have added the following clarification to the text (P3, L15-26): "While admittedly somewhat subjective, this 50 cm$^{-3}$ threshold is consistent with the upper bound of near-surface and below-cloud observations in MBL environments routinely featuring exceptionally low $N_A$ such as subtropical pockets of open cells (Abel et al., 2019; Sharon et al., 2006; Terai et al., 2014), mid-latitude open-cellular convection (Abel et al., 2017; Pennypacker and Wood, 2017) and across the trade wind stratocumulus-to-cumulus transition (Bretherton et al., 2019). It is also well situated within the typical range ($\sim$30 – 60 cm$^{-3}$) of number concentrations used for the lowest aerosol cases in large eddy simulation studies of MBL aerosol-cloud interactions (Wang et al., 2010; Wang and Feingold, 2009; Yamaguchi and Feingold, 2015; Zhou et al., 2017). Prior work defined ultra-clean layers near the top of the MBL, often observed in the stratocumulus-to-cumulus transition, with $N_A < 10$ cm$^{-3}$ (Wood et al., 2018). We argue it is reasonable to set a higher threshold near the surface, where aerosol number concentrations are generally higher due to proximity to the sea spray source. Furthermore, Wood et al. (2018) focused on these layers primarily as a mesoscale feature within larger cloud systems, whereas our interest is in studying ultra-clean conditions as daily-scale events. Defining ultra-clean conditions using daily median $N_A < 50$ cm$^{-3}$ balances the need to reasonably capture conditions with exceptionally low near-surface $N_A$ in the remote MBL while maintaining a robust sample of cases to study."

P.3 line 30: Would high aerosol counts but low CO/rBC qualify as 'polluted' ? The authors suggest this might occur during February. Overall a bit more description of the high-Na days would be helpful. Are they all from the months when smoke is clearly present?

Thank you for raising the need to clarify this point. The polluted back trajectories are only from the months where there are also ultra-clean days (i.e. during the biomass burning season). We have added text in the Abstract (P1, L16) and Section 2.2 (P4, L20-21), where the back trajectories are introduced, to ensure that this is clear. We have also added a supplemental table (Table S1) listing all of the ultra-clean and polluted dates from those same months that are included in the analysis.

Section 3.2, fig. 3: It would be interesting to also discriminate further those days that are more truly pristine. Do those correspond to the back-trajectories that more clearly go back to the southern oceans? There may not be many days with daily median CO values <~ 60 ppb and rBC values within the sensitivity limit, but there should be some, and it would add interest to hear about those as well.

We have expanded Section 3.2 (P7, L4-9), including the addition of a new supplemental figure, with the following: "However, trajectory latitude at seven days back from ASI only explains 25% of the variance in daily median CO concentrations across ultra-clean days. Trajectories from days with daily median CO $\leq$ 59 ppb (n = 6), the non-burning background median concentration, can be anywhere between 40° - 60°S at seven days back from ASI (Figure S2). Overall, boundary layer air mass origin is a relatively weak predictor of downwind variability in CO concentration on ultra-clean days." So while there is a weak suggestion that if a boundary layer trajectory originates farther toward the mid-latitudes/the Southern Ocean it ends up with lower CO around ASI, this is still not the dominant driver of variability across ultra-clean days.

This continues to point to the importance of variability above the boundary layer rather than to some systematic difference in boundary layer origin and/or trajectory.

*Figures:*

Fig. 1: It's hard to tell how many UC days occur per month from panel a and b. One idea would be to mention how many occur each month near the top of the figure.
In response to this and other reviewer comments, we have added a supplemental table (Table S1) that lists all the ultra-clean days for reference.

Fig. 2 panels c and d: I suppose this is saying something about temporal variability as well, with hourly values being shown for a given daily median threshold on Na. For completeness it would be nice to see a similar plot for the pdf of the hourly median Na. It would be a fifth panel. Not sure what to suggest for a 6th panel to balance it visually.
We had previously examined hourly $N_A$ on ultra-clean days as a consistency check on the representativeness of the daily median concentrations (see box plots below for data by month, for example). There is certainly finer scale variability in $N_A$ within ultra-clean days, but this is not the focus of our analysis. Figure 2 highlights the burning tracer results that more directly inform the conclusions of the paper.

[Figure]

Fig. 4: the cumulative distributions take some study to interpret. Have the authors considered a normalized frequency distribution instead? Same for Fig. 5a.
We appreciate that CDF and PDF plots can both be useful. Our main goal here is to highlight the general differences and shifts in the distributions, which the CDF plots nicely demonstrate. To us, they also more clearly represent the parts of the distribution that contribute most to those shifts and by how much.

Fig. 6: I don't see a clear correspondence between LWP and UC days through this figure. I wonder if the MWR LWP data are simply too local.
Figure 6 shows the monthly boxplot distributions and seasonality for all of the LASIC TSI cloud fraction and MWR LWP data. This provides context for our observations, but is not just for any particular period featuring ultra-clean days. Ultra-clean days are only observed in months around the seasonal peak in cloud cover and LWP in this local dataset. The enhancements in coalescence scavenging, which would require cloudiness and high LWP, that we argue are likely responsible for driving ultra-clean conditions observed at ASI occur against the backdrop this broader seasonal pattern. These local observations will certainly be more variable than those over an

entire region or with a longer temporal record. We agree that local cloud properties will not be the only factor in driving ultra-clean conditions, and we point to the need to consider the detailed Lagrangian evolution of these boundary layers over several days in future work. We also reference prior work with satellite observations from the region that are largely consistent with the smaller-scale ASI observations (O'Dell et al., 2008; Zuidema et al., 2016). Nevertheless, even the local observations available from LASIC are consistent with a shift in cloud & precipitation properties (situated around seasonal maxima in cloud fraction and LWP) that could drive and maintain strong aerosol loss rates on ultra-clean days.

O'Dell, C. W., Wentz, F. J. and Bennartz, R.: Cloud liquid water path from satellite-based passive microwave observations: A new climatology over the global oceans, J. Clim., 21, 1721–1739, doi:10.1175/2007JCLI1958.1, 2008.

Zuidema, P., Chang, P., Medeiros, B., Kirtman, B. P., Mechoso, R., Schneider, E. K., Toniazzo, T., Richter, I., Small, R. J., Bellomo, K., Brandt, P., De Szoeke, S., Farrar, J. T., Jung, E., Kato, S., Li, M., Patricola, C., Wang, Z., Wood, R. and Xu, Z.: Challenges and prospects for reducing coupled climate model SST biases in the eastern tropical atlantic and pacific oceans: The U.S. Clivar eastern tropical oceans synthesis working group., 2016.

**Referee #4**

*Manuscript Summary*
The authors analyze surface observations of aerosol, gas phase composition, and cloud properties at Ascension Island over a period of 16 months, acquired during the Layered Atlantic Smoke Interactions with Clouds (LASIC) campaign. Back-trajectory calculations support the analysis. The authors distinguish three aerosol states at Ascension Island: Background conditions, polluted conditions, and ultraclean conditions. Ultraclean conditions are defined based on a daily median concentration of aerosol particles (CCN) with dry diameters between 60 nm and 1 μm < 50 cm−3 . The authors find 41 days with ultraclean conditions at Ascension Island. All of these occur during the South-West African biomass burning season. A portion of the ultraclean days also exhibits carbon monoxide and refractory black carbon levels above background. Apart from ultraclean days, boundary layer CCN concentrations at Ascension Island are significantly elevated above background levels. No days with ultraclean conditions are found outside the biomass burning season, which defines background conditions. The authors conclude, based on analysis of carbon monoxide and refractory black carbon levels, statistics of precipitation and liquid water path at Ascension Island, and back-trajectory calculations that CCN concentrations are low on the ultraclean days not because originally clean air has been advected to Ascension Island, but because enhanced coalescence scavenging in low clouds has strongly reduced CCN in polluted air masses. This is an interesting result because it points to a more complex interaction between (anthropogenic) aerosol and cloud properties in the region, with causal links in both directions.

*Review Summary*
In their analysis of the observations the authors accumulated a good amount of circumstantial evidence to render their hypothesis plausible, although the analyzed data are specific to conditions at Ascension Island only and hence do not establish a causal connection between conditions on ultraclean days and processes that may give rise to them. Although not quantitative, the back-trajectory analysis is helpful. The study is, as the authors point out in their closing statements, a good motivation and starting point for subsequent investigations. There are a few points that I would ask the authors to look after, listed below. Otherwise, the manuscript is in good shape.

Major comments –

Could there be other explanations for the ultraclean days than enhanced coalescence scavenging in low clouds with higher liquid water content? E.g., is it possible that on the ultraclean days, the polluted air has entrained earlier into the boundary layer, hence spent a more time there compared to other days during the biomass burning season? A longer sojourn in the boundary layer would give coalescence scavenging more time to deplete the aerosol. Please comment and if applicable, discuss in the manuscript.
We agree that the time spent experiencing coalescence scavenging could also play a role in $N_A$ variability sampled at ASI. Addressing when particular air masses entrained into the MBL wasn't the focus of this analysis since we are studying isobaric boundary layer trajectories. However, we have previously examined standard 3D trajectories (only for 2016) and did not find

an immediately apparent systematic difference in the timing of when the ultra-clean day (black) and polluted (red) trajectories crossed below a 1.5 km threshold taken as roughly indicative of MBL height in the region. There is a fair amount of spread in height across the trajectories 3-5 days back from ASI, but both the trajectory and cloud/precipitation results presented still point to the importance of sources (smoke structure above the MBL) and sinks (scavenging) encountered along the way in driving the bulk of variability downwind rather than any consistent differences in the trajectories themselves. For future work, we do note the importance of considering the detailed evolution of a range of these cases in a Lagrangian modeling framework in the Discussion. This will hopefully provide more insight into the relative importance of different processes and their associated time scales along trajectories that cannot be addressed with the observations here.

[Figure]

Please calculate the average speed of the trajectories between 35 S and Ascension Island. Is there a difference in advection velocity between the ultraclean and nonultraclean days during the biomass burning season? If yes, discuss what this could mean for the processes that cause ultraclean conditions.
Thank you for raising this possibility. We do not observe any systematic differences in advection velocity between ultra-clean and polluted trajectories. Almost all boundary trajectories converge to within 2-3 lat/lon degrees of each other approximately 3 days back, leading to minimal differences in the speed with which they approach ASI. The large-scale horizontal circulation in the boundary layer is largely consistent once in the SEATL trades. We have now made note of this fact in the text (P7, L10-11).

The criterion for what makes ultraclean conditions varies between works. Albrecht et al. (doi:10.1175/BAMS-D-17-0180.1), e.g., define ultraclean conditions as having aerosol concentrations of less than 10 cm−3 in the nominal range between 0.06 - 1.0 μm., while in the present work it is < 50 cm−3 . Please add a passage mentioning the different criteria and explain why in the present work the criterion of < 50 cm−3 was chosen.
In response to this and other similar reviewer comments, we have added the following clarification to the text (P3, L15-26): "While admittedly somewhat subjective, this 50 cm$^{-3}$ threshold is consistent with the upper bound of near-surface and below-cloud observations in MBL environments routinely featuring exceptionally low $N_A$ such as subtropical pockets of open cells (Abel et al., 2019; Sharon et al., 2006; Terai et al., 2014), mid-latitude open-cellular convection (Abel et al., 2017; Pennypacker and Wood, 2017) and across the trade wind stratocumulus-to-cumulus transition (Bretherton et al., 2019). It is also well situated within the typical range (~30 – 60 cm$^{-3}$) of number concentrations used for the lowest aerosol cases in large eddy simulation studies of MBL aerosol-cloud interactions (Wang et al., 2010; Wang and

Feingold, 2009; Yamaguchi and Feingold, 2015; Zhou et al., 2017). Prior work defined ultra-clean layers near the top of the MBL, often observed in the stratocumulus-to-cumulus transition, with $N_A < 10$ cm$^{-3}$ (Wood et al., 2018). We argue it is reasonable to set a higher threshold near the surface, where aerosol number concentrations are generally higher due to proximity to the sea spray source. Furthermore, Wood et al. (2018) focused on these layers primarily as a mesoscale feature within larger cloud systems, whereas our interest is in studying ultra-clean conditions as daily-scale events. Defining ultra-clean conditions using daily median $N_A < 50$ cm$^{-3}$ balances the need to reasonably capture conditions with exceptionally low near-surface $N_A$ in the remote MBL while maintaining a robust sample of cases to study."

How robust is the number of ultraclean days to the UHSAS < 50 cm−3 criterion?
See the comment above for our discussion of how we arrived at the threshold of 50 cm$^{-3}$. If you change the cutoff, you would of course increase or decrease the number of days in your sample (60 cm$^{-3}$ puts you at 56 days, 30 cm$^{-3}$ puts you at 19) but either start including conditions that deviate more from the observations and modeling of MBL environments typically characterized by exceptionally low $N_A$ or limit the number of available days to study. The updated text above includes a consideration of that balance (see comment above).

"... with the correlation statistically indistinguishable from zero (r2 = 0.06), ..." To make this statement you /must/ calculate the p-value of the linear regression/correlation. Without the p-value, there is no way of telling whether a correlation coefficient/coefficient of determination is statistically indistinguishable from zero, regardless of its numerical value.
Thank you for raising the need to clarify this point. Our statement that this correlation ($N_A$ vs. $N_{CN3}$ on ultra-clean days) is statistically indistinguishable from zero is based on the fact that the correlation coefficient's calculated 95% confidence interval includes zero. We have updated the text to make this clear (P5, L27-28): "This relationship is substantially weaker ($r^2 = 0.06$), with the 95% confidence interval for this correlation including zero, on ultra-clean days (Figure 1d)."

*Other comments*

Please check the text for sentences that can be simplified; some are hard to understand. For example, "The relative invariance of isobaric boundary layer back trajectories between ultraclean and the most polluted days at ASI suggests that the potential for BBA entrainment set by the vertical separation of a smoke layer and the evolution of the boundary layer cloud field plays a larger role in upwind (e.g. at ASI) aerosol variability than a systematic difference in large-scale horizontal circulation in the boundary layer." is rather difficult to decipher.
Thank you for the suggestion to clear up the language here, we have separated this into two more straightforward sentences.

Please mention the meteorological input that you used to drive the HYSPLIT model.
Thank you for pointing out the need to include this information. The text has been updated to address this point: "We take a complimentary approach by analysing 7-day isobaric boundary layer back-trajectories initialized at approximately 500 m over ASI at 12:00 UTC as computed by the NOAA Hybrid Single Particle Lagrangian Integrated Trajectory Model (HYSPLIT) with Global Data Assimilation System meteorology on a 0.5 degree by 0.5 degree grid (Stein et al., 2015)."

500 m trajectories are not isobaric.

In the updated text, we further clarify that the trajectories are *initialized* at approximately 500 m over ASI.

Please consider if the labeling of the abscissa in the plots that show data as a function of the month is precise enough to inform the reader on the actual point in time (are the vertical lines the 1st of the month or the 15th?)

The full time series (Figures 1a,b; 2a,b) are intended to give a general sense of how ultra-clean days fit into the larger pattern and seasonal cycle of aerosol and trace-gas observations at ASI, rather than for tracking the details around any one event or even month.

- Furthermore, in response to several reviewer comments we have added a supplemental table (Table S1) that lists all of the ultra-clean (and polluted) days for reference so individual events can be identified quickly if needed.
- We have updated the captions in Figures 1 and 2 to clarify that the vertical grid lines mark the first of each month labeled on the tick.

[revised manuscript text omitted]

**Author**
**Comment [8]:** Added in response to comments from Reviewer #2.

**Author**
**Comment [9]:** Added in response to comments from Reviewer #2.

**Author**
**Comment [10]:** Edited to include details of meteorological forcing in response to comments from Reviewers #2 and #4.

**Author**
**Comment [11]:** Added in response to comments from Reviewers #2 and #3

**Author**
**Comment [12]:** Added in response to comments from Reviewer #2.

[revised manuscript text omitted]

There is also some overlap in the distributions of ultra-clean and the non-burning background SP2 rBC (Dec. 2016, March-April 2017, Figure 2d). However, as with CO, the statistics do indicate a shift toward overall higher concentrations on ultra-clean days. The median of hourly median SP2 rBC is 51 ng m$^{-3}$ with an inter-quartile range of 23 - 120 ng m$^{-3}$ on ultra-clean days, compared to the background median of 20 ng m$^{-3}$ and inter-quartile range of 12 - 45 ng m$^{-3}$. Even the hourly extremes captured by the 5th and 95th percentiles are higher on ultra-clean days (12 and 312 ng m$^{-3}$) than across the non-burning background (10 and 135 ng m$^{-3}$). In summary, there is no indication that ultra-clean days are devoid of BBA signatures or even exhibit the same distribution of smoke tracer concentrations as the non-burning season background at ASI. We will return to the implication of these results for the characterization of extremely low aerosol number events as 'ultra-clean' in the Discussion.

**Author**
**Comment [16]:** Months added in response to comments from Reviewer #2.

**Author**
**Comment [17]:** Edited in response to comments from Reviewer #2.

**Author**
**Comment [18]:** Added in response to comments from Reviewer #2.

**Author**
**Comment [19]:** Edited in response to comments from Reviewer #2.

**Author**
**Comment [20]:** Added in response to comments from Reviewer #3.

Relative to the polluted extremes (recall these are defined by daily median $N_A$ above monthly 95th percentile), there are somewhat more ultra-clean boundary layer isobaric back-trajectories that originate farther toward the mid-latitudes and the Southern Ocean (Figure 3a). We might expect lower background aerosol concentrations and weaker influence from African biomass burning in these air masses than in those spending more time in the subtropics, helping explain the subset of ultra-clean days with burning tracer concentrations closer to background levels. However, trajectory latitude at seven days back from ASI only explains 25% of the variance in daily median CO concentrations across ultra-clean days. Trajectories from days with daily median CO $\leq$ 59 ppb (n = 6), the non-burning background median concentration, can be anywhere between 40° - 60°S at seven days back from ASI (Figure S2). Overall, boundary layer trajectory origin is a relatively weak predictor of downwind variability in CO concentration on ultra-clean days. Furthermore, there are many polluted and ultra-clean boundary layers that follow similar isobaric trajectories on their way toward ASI (Figure 3b). By three days away from ASI, most trajectories have converged to within three to four degrees latitude and longitude of each other. In other words, the boundary layers that would be entraining smoke from the tree troposphere often follow very similar horizontal circulation patterns for both the highest and lowest upstream extremes of $N_A$. This all points to a smaller role for variations in large-scale horizontal circulation in the SEATL MBL in driving aerosol and trace gas variability observed at ASI.

**3.3 Precipitation and Cloud Liquid Water**

Ultra-clean days exhibit markedly different surface precipitation characteristics, as measured by the ASI Parsivel2. The distribution of precipitation rates shifts toward higher intensities on ultra-clean days (Figure 4a). Precipitation is also much more common on ultra-clean days (Figure 4b), with almost 90% of non-UC days having a precipitation frequency of less than 0.05. The tendency for more frequent and more intense precipitation inevitably leads to higher total accumulation on ultra-clean days (Figure 4c). The difference mostly stems from the shift toward more frequent drizzle conditions in ultra-clean conditions. These data are all presented with cumulative distributions in order to concisely highlight the generally different behaviour of precipitation across ultra-clean days. However, the increase in drizzle intensity, frequency and accumulation also holds for ultra-clean days relative only to the distribution within their respective months (not shown).

The median LWP retrieved by MWR measurements is higher on ultra-clean days (110 g m$^{-2}$) compared to other days (76 g m$^{-2}$). The inter-quartile spreads are actually larger than the median LWP whether within ultra-clean days (41 – 235 g m$^{-2}$) or not (26 – 192 g m$^{-2}$). These statistics are further illustrated by the difference in the LWP cumulative distributions (Figure 5a). The shift is noted across most of the sampled range of LWP, though the distributions do overlap at the very highest values. While the shift toward higher LWP on ultra-clean days may not appear substantial, recall that coalescence scavenging is non-linearly dependent on LWP (Wood, 2006). The approximately 35% increase in median LWP on ultra-clean days would strengthen the coalescence scavenging aerosol sink by 70%.

**Comment [21]:** Added in response to comments from Reviewer #3.

**Comment [22]:** Added in response to comments from Reviewer #4

**Comment [23]:** Edited to include medians and inter-quartile ranges, rather than means, in response to comments from Reviewer #2.

[revised manuscript text omitted]

Author
**Comment [25]:** Added in response to comments from Reviewer #3

Author
**Comment [26]:** Sentences separated and edited in response to comments from Reviewer #4.

[revised manuscript text omitted]

Author
Comment [28]: All figures have been updated with thicker grid spacing and more spacing for the tick labels to improve readability.

Author
Comment [29]: $r^2$ values added in response to comments from Reviewer #2.

[Figure]

**Figure 2:** Time series of daily and monthly median (a) CO and (b) rBC measured during LASIC, with ultra-clean days marked in cyan. In (a) and (b), vertical grid lines mark the first of each month labelled on the tick. We then compare the PDF of hourly median (c) CO and (d) rBC from ultra-clean days to the PDF of hourly median concentrations from each tracers' respective non-burning background.

**Author**

**Comment [30]:** Figure updated to include PDFs in response to comments from Reviewer #1.

**Isobaric Back Trajectories**

(a) Ultra-clean, boundary layer

[Figure]

(b) Polluted, boundary layer

[Figure]

**Figure 3:** Isobaric 7-day HYSPLIT back trajectories at 500 m for (a) ultra-clean and (b) polluted

[Figure]

[Figure]

[Figure]

**Figure 4:** Cumulative distributions of (a) instantaneous precipitation rate, (b) daily precipitation frequency and (c) daily precipitation accumulation as measured by the ASI Parsivel2 laser disdrometer.

[Figure]

[Figure]

**Figure 5:** (a) Comparison of the cumulative distributions of best-estimate LWP retrieval from the ASI MWR between ultra-clean and all other days and (b) medians/standard deviations of daily median MWR LWP across bins of daily median accumulation mode aerosol for the entire LASIC record. In (b), bin widths were selected to account for varying density of days across the range of aerosol concentrations while still visualizing the broader pattern.

[Figure]

**Figure 6:** Monthly boxplots of (a) daily mean Total Sky Imager cloud fraction and (b) daily median MWR best-estimate LWP for each month in the LASIC record.